# An Early Warning Sign of Critical Transition in The Antarctic Ice Sheet - A Data Driven Tool for Spatiotemporal Tipping Point

Abd AlRahman AlMomani[1,2] and Erik Bollt[1,2]

[1]Department of Electrical and Computer Engineering, Clarkson University, Potsdam, NY 13699, USA
[2]Clarkson Center for Complex Systems Science ($C^3S^2$), Potsdam, NY 13699, USA

**Correspondence:** Abd AlRahman AlMomani (aaalmoma@clarkson.edu)

**Abstract.** Our recently developed tool, called Directed Affinity Segmentation was originally designed for data-driven discovery of coherent sets in fluidic systems. Here we interpret that it can also be used to indicate early warning signs of critical transitions in ice shelves as seen from remote sensing data. We apply a directed spectral clustering methodology, including an asymmetric affinity matrix and the associated directed graph Laplacian, to reprocess the ice velocity data and remote sensing satellite images of the Larsen C ice shelf. Our tool has enabled the simulated prediction of historical events from historical data, fault lines responsible for the critical transitions leading to the break up of the Larsen C ice shelf crack, which resulted in the A68 iceberg. Such benchmarking of methods using data from the past to forecast events that are now also in the past is sometimes called post-casting, analogous to forecasting into the future. Our method indicated the coming crisis months before the actual occurrence.

## 1  Introduction

Warming associated with climate change causes the global sea level to rise Mengel et al. (2016). Three primary reasons for this are ocean expansion McKay et al. (2011), ice sheets losing ice faster than it forms from snowfall, and glaciers at higher altitudes melting. During the $20^{th}$ century, sea level rise has been dominated by glaciers' retreat. This has started to change in the $21^{st}$ century because of the increased iceberg calving. Seroussi et al. (2020); Mengel et al. (2016) Ice sheets store most of the land ice (99.5%) Mengel et al. (2016), with a sea-level equivalent (SLE) of 7.4m for Greenland and 58.3m for Antarctica. Ice sheets form in areas where the snow that falls in winter does not melt entirely over the summer. Over thousands of years of this effect, the layers grow thicker and denser as the weight of new snow and ice layers compresses the older layers. Ice sheets are always in motion, slowly flowing downhill under their weight. Much of the ice moves through relatively fast-moving outlets called ice streams, glaciers, and ice shelves near the coast. When a marine ice sheet accumulates a mass of snow and ice at the same rate as it loses mass to the sea, it remains stable. Antarctica has already experienced dramatic warming. Especially, the Antarctic Peninsula, which juts out into relatively warmer waters north of Antarctica, which has warmed 2.5 degrees Celsius (4.5 degrees Fahrenheit) since 1950  NASA (2017).

A large area of the Western Antarctic Ice Sheet is also losing mass, attributed to warmer water up-welling from the deeper ocean near the Antarctic coast. In Eastern Antarctica, no clear trend has emerged, although some stations report slight cooling.

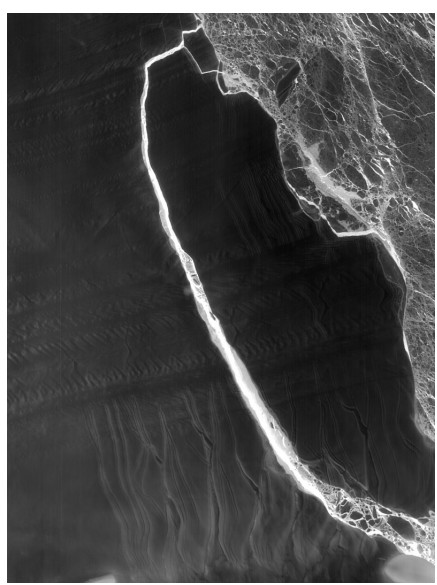

**Figure 1.** A-68 iceberg. The fractured berg and shelf are visible in these images, acquired on July 21, 2017, by the Thermal Infrared Sensor (TIRS) on the Landsat 8 satellite. Credit: NASA Earth Observatory images by Jesse Allen, using Landsat data from the U.S. Geological Survey.

Overall, scientists believe that Antarctica is starting to lose ice  NASA (2017), but so far, the process is not considered relatively fast as compared to the widespread changes in Greenland  NASA (2017).

Since 1957, the continent-wide average's current record reveals a surface temperature trend of Antarctica that has been positive and significant at $> 0.05 °$ C/decade Steig et al. (2009); Gagne et al. (2015). Western Antarctica has warmed by more than 0.1 °C/decade in the last 50 years, and this warming is most active during the winter and spring. Although this is partly 30 offset by autumn cooling in Eastern Antarctica, this effect was prevalent in the 1980s and 1990s Steig et al. (2009).

Of particular interest to us in this presentation, the Larsen Ice Shelf extends like a ribbon down from the East Coast of the Antarctic Peninsula, from James Ross Island to the Ronne Ice Shelf. It consists of several distinct ice shelves, separated by headlands. The major Larsen C ice crack was already noted to have started in 2010 Jansen et al. (2015). Still, it was initially very slowly evolving, and there were no signs of radical changes according to Interferometry studies of the remote sensing 35 imagery Jansen et al. (2010b). However, since October 2015, the major ice crack of Larsen C had been growing more quickly, until the point where recently it finally failed, resulting in calving the massive A68 iceberg. See Fig. 1; this is the largest known iceberg, with an area of more than 2,000 square miles, or nearly the size of Delaware. In summary, A68 detached from one of the largest floating ice shelves in Antarctica and floated off into the Weddell Sea.

In Glasser et al. (2009), the authors presented a structural glaciological description of the system and subsequent analysis of 40 surface morphological features of the Larsen C ice shelf, as seen from satellite images spanning the period 1963–2007. Their research results and conclusions stated that: "*Surface velocity data integrated from the grounding line to the calving front along*

*a central flow line of the ice shelf indicate that the residence time of ice (ignoring basal melt and surface accumulation) is 560 years. Based on the distribution of ice-shelf structures and their change over time, we infer that the ice shelf is likely to be a relatively stable feature and that it has existed in its present configuration for at least this length of time.".*

In Jansen et al. (2010a), the authors modeled the flow of the Larsen C and northernmost Larsen D ice shelves using a model of continuum mechanics of the ice flow. They applied a fracture criterion to the simulated velocities to investigate the ice shelf's stability. The conclusion of that analysis shows the Larsen C ice shelf is inferred to be stable in its current dynamic regime. This work was published in 2010. According to analytic studies, the Larsen C ice crack already existed at that time, but was considered slowly-growing. There was no expectations at that time that the crack growth would proceed quickly and

that collapse of the Larsen C was imminent.

    Interferometry has traditionally been the primary technique to analyze and predict ice cracks based on remote sensing. Interferometry Bassan (2014); Lämmerzahl et al. (2001) constitutes a family of techniques in which waves, usually electromagnetic waves, are superimposed, causing the phenomenon of interference patterns, which in turn are used to *"extract information"* concerning the underlying viewed materials. Interferometers are widely used across science and industry to measure small dis-

placements, refractive index changes, and surface irregularities. So it is considered a robust and familiar tool that is successful in the macro-scale application for monitoring the structural health of the ice shelves. Here we will instead take a data-driven approach, directly from the remote sensing imagery, to infer structural changes indicating the impending tipping point toward Larsen C's critical transition, and eventual breakup.

    Fig. A1 shows the interferometry image as of April 20, 2017. Although it clearly shows the crack that already existed at that

time, but this case, apparently it provided no information concerning forecasting the breakup that soon followed. Just a couple of weeks after the image shown in Fig. A1, the Larsen C ice crack changed significantly and presented a different dynamic that quickly divided into two branches, as shown in Fig. A2. Interferometry is a powerful tool for detecting spatial variations of the ice surface velocity. However, when it comes to inferring early stages of future critical transitions, it did not provide useful indications portending the important event that soon followed. Therefore, there is a clearly need for other methods that may

be capable of this task. As we will show, our method achieves a very useful and successful data-driven early indicator of this important outcome.

## 2   Directed Partitioning

In our previous work Al Momani (2017); AlMomani and Bollt (2018), we developed the method of Directed Affinity Segmentation (DAS), and we showed that our method is a data-driven analogue to the transfer operator formalism designed. DAS was

originally designed to characterize coherent structures in fluidic systems, such as ocean flows or atmospheric storms. Further, DAS is truly a data-driven method in that it is suitable even when these systems are observed only from movie data and specifically without either an exact differential equation, nor the need for the intermediate stage of modelling the vector field Luttman et al. (2013), responsible for underlying advection. In the current work, we apply this concept of seeking coherent structures

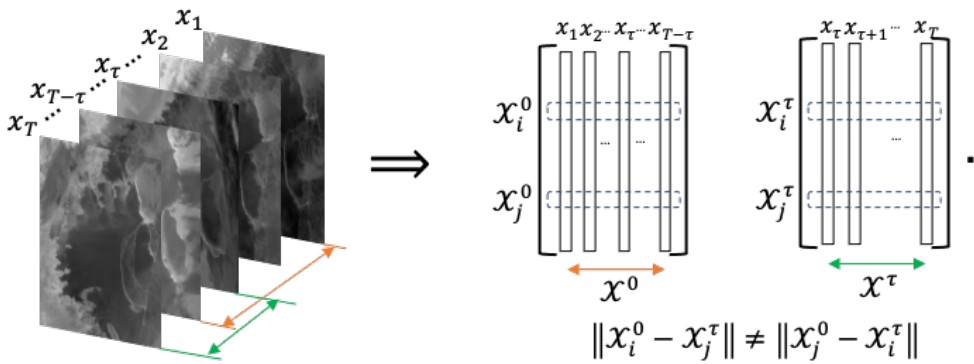

**Figure 2.** Directed partitioning method. We see the image sequence to the left, and to the right, we reshape each image as a single column vector. Following the resultant trajectories, we see that the pairwise distance between the two matrices will result in an asymmetric matrix. Raw images source Scambos et al. (1996).

under the hypothesis that a large ice sheet that begins to move in mass, appears a great deal like a mass of material in a fluid that holds together in what is often called a coherent set.

Two of the most commonly used and successful image segmentation methods are based on 1) $k$-means Kanungo et al. (2002), and 2) spectral segmentation Ng et al. (2002), respectively. However, while these were developed successfully for static images, they require major adjustments for successful application to sequences of images, i.e., movies. The spatiotemporal problem of motion segmentation is associated with coherence, despite that traditionally they are considered well suited to static images Shi and Malik (2000). The key difference between image segmentation of static images, and coherence as related to motion segmentation is what underlies a notion of coherent observations, since we must also consider the directionality of the arrow of time.

Defining a loss function of some kind is often the starting point when specifying an algorithm in machine learning. An affinity measure is the phrase used to describe a comparison, or cost, between states. In this case, a state may be the measured attributes at a given location in an image scene. However, when there is an underlying arrow of time, the loss functions that most naturally arise to track coherence will not be inherently symmetric. Correspondingly, affinity matrices associate the affinity measure for each pairwise comparison across a finite data set. A graph is associated with the affinity matrix, where there is an edge between each state for which there is a nonzero affinity. Generally, in the symmetric case, these graphs are undirected. Now consider that if the affinity matrices are not symmetric, then these are associated with *directed graphs,* which describes the arrow of time. This is a theoretical complication to standard methodology since much of the theoretical underpinnings of standard spectral partitioning assumes a symmetric matrix corresponding to an undirected graph and then considers the spectrum of eigenvalues of the corresponding symmetric graph Laplacian matrix that follows. This new case can be accommodated by spectral graph theory, as there is a graph Laplacian for weighted directed graphs, built upon the theoretical work of F. Chung Chung and Oden (2000). Our own work in AlMomani and Bollt (2018), specialized this concept of directed spectral graph theory, to the scenario of image sequences derived from an assumed underlying evolution operator.

To proceed with our directed partitioning method, we formulate the (movie) imagry sequences data set as the following matrices;

$$\mathcal{X}^0 = [X_1 | X_2 | ... | X_{T-\tau}], \tag{1}$$

$$\mathcal{X}^\tau = [X_{\tau+1} | X_{\tau+2} | ... | X_T], \tag{2}$$

where each $X_i$ is the $i^{th}$ image (or the image at $i^{th}$ time step) describes a $d_1 \times d_2$ pixelated image reshaped as a column vector $d \times 1$, $d = d_1 d_2$. See Fig. 2. This describes a gray-scale image, but in the likely scenario of multiple attributes or color bands at each pixel, then likewise these data structures include the corresponding tensor depth. Here, $\tau$ is the time delay, $\mathcal{X}_0$ and $\mathcal{X}_\tau$ are the images sequences stacked as column vectors with a time delay at the current and future times respectively. Choosing the value of the time delay $\tau$, can results in significant differences in the segmentation process. Consider that in the case of a relatively slowly evolving dynamical system, where the change between two consecutive images is not significantly distinguishable, then choosing a large value for $\tau$ may be better suited. In our work, we considered the mean image over a period of one-month as a moving window generates our images, which implies $\tau$ to be one month.

Note that the rows of $\mathcal{X}^0, \mathcal{X}^\tau \in \mathbb{R}^{d \times T-\tau}$ represent the change of the color of the pixel at a fixed spatial location $z_i$. It is crucial to keep in mind that we chose the color as the evolving quantity for a designated spatial location for clarity and consistency with our primary application and approach described in this paper. However, we can select the evolving quantity to be the magnitude of the pixels obtained from spectral imaging, or experimental measures obtained from the field, such as pressure, density, or velocity. The results section introduces examples where the ice surface velocity was used instead of the color to highlight how results may vary based on the selected measure.

We introduced AlMomani and Bollt (2018) an affinity matrix in terms of a pairwise distance function between the pixels $i$ and $j$ as,

$$D_{i,j} = \mathcal{S}(\mathcal{X}_i^0, \mathcal{X}_j^\tau) + \alpha \mathcal{C}(\mathcal{X}_i^0, \mathcal{X}_j^\tau, \tau) \tag{3}$$

where the function $\mathcal{S} : \mathbb{R}^2 \mapsto \mathbb{R}$ is used to define the spatial distance between pixels $i$ and $j$ describing physical locations $z_i$ and $z_j$. The function $\mathcal{C} : \mathbb{R}^{T-\tau} \times \mathbb{R}^{T-\tau} \times \mathbb{R} \mapsto \mathbb{R}$ is a distance function describing "color distance" between the $i^{th}$ and the $j^{th}$ color channels. The parameter $\alpha \geq 0$ regularizes, balancing these two effects. The value of $\alpha$ can be seen as a degree of importance of the function $\mathcal{C}$ relative to the spatial change. Large values of $\alpha$ will make the color variability dominate the distance in Eq. 3, and it would classify "very" close (spatially) regions as different coherent sets when they have small color differences. On the other hand, small values of $\alpha$ may classify spatially neighboring regions as one coherent set, even when they have a significant color difference. In our work, the color is quantified as a gray-scale color of the images ($\mathcal{C} \in [0, 1]$). So, we scaled the value of $\mathcal{S}$ to be in $[0, 1]$, then we choose $\alpha = 0.25$, to emphasize spatial change, where we choose the functions $\mathcal{S}$ and $\mathcal{C}$ each to be $L_2$-distance functions,

$$\mathcal{S}z_i, z_j) = \|z_i - z_j\|_2, \tag{4}$$

and

$$\mathcal{C}(\mathcal{X}_i^0, \mathcal{X}_j^\tau, \tau) = \|\mathcal{X}_i^0 - \mathcal{X}_j^\tau\|_2. \tag{5}$$

We see that the spatial distance matrix $\mathcal{S}$ is symmetric. However, the color distance matrix $\mathcal{C}$ is asymmetric for all $\tau > 0$. While the matrix generated by $\mathcal{C}(\mathcal{X}_i^0, \mathcal{X}_j^\tau, 0)$ refers to the symmetric case of spectral clustering approaches, we see that the matrix given by $\mathcal{C}(\mathcal{X}_i^0, \mathcal{X}_j^\tau, \tau)$, $\tau > 0$ implies an asymmetric cost naturally due to the directionality of the arrow of time. Thus we require that an asymmetric clustering approach must be adopted.

First we define our affinity matrix from Eq. 3 as,

$$\mathcal{W}_{i,j} = e^{-D_{i,j}^2/2\sigma^2}. \tag{6}$$

This has the effect that both spatial and measured (color) effects have "almost" Markov properties, as far field effects are almost "forgotten" in the sense that they are almost zero. Likewise, near field values are largest. Notice that we have suppressed including all the parameters in writing $\mathcal{W}_{i,j}$, including time parameter $\tau$ that describes sampling history, and the parameters $\alpha$ and $\sigma$ that serve to balance spatial scale and resolution of color histories.

We proceed to cluster the spatiotemporal regions of the system, in terms of the directed affinity $\mathcal{W}$ by interpreting the problem as random walks through the weighted *directed* graph, $G = (V, E)$ designed by $\mathcal{W}$ as a weighted adjacency matrix. Let,

$$\mathcal{P} = \mathcal{D}^{-1}\mathcal{W}, \tag{7}$$

where

$$\mathcal{D}_{i,j} = \begin{cases} \sum_k \mathcal{W}_{i,k}, & i = j, \\ 0, & i \neq j, \end{cases} \tag{8}$$

is the degree matrix, and $\mathcal{P}$ is a row stochastic matrix representing probabilities of a Markov chain through the directed graph $G$. Note that $\mathcal{P}$ is row stochastic implies that it row sums to one. This is equivalently stated that the right eigenvector is the ones vector, $\mathcal{P}\mathbf{1} = \mathbf{1}$, but the left eigenvector corresponding to left eigenvalue 1 represents the steady state row vector of the long term distribution,

$$u = u\mathcal{P}. \tag{9}$$

Consider for example, if $\mathcal{P}$ is irreducible, then $u = (u_1, u_2, ..., u_{pq})$ has all positive entries, $u_j > 0$ for all $j$, or said for simplicity of notation, $u > 0$, interpreted componentwise. Let $\Pi$ be the corresponding diagonal matrix,

$$\Pi = diag(u), \tag{10}$$

and likewise,

$$\Pi^{\pm 1/2} = diag(u^{\pm 1/2}) = diag((u_1^{\pm 1/2}, u_2^{\pm 1/2}, ..., u_{pq}^{\pm 1/2})), \tag{11}$$

which is well defined for either $\pm$ sign branch when $u > 0$.

Then, we may cluster the directed graph using spectral graph theory methods specialized for directed graphs, following the weighted directed graph Laplacian described by Fan Chung Chung (2005). A similar computation has been used for transfer operators in Froyland and Padberg (2009); Hadjighasem et al. (2016) and as reviewed Bollt and Santitissadeekorn (2013); Santitissadeekorn and Bollt (2007); Bollt et al. (2012), including in oceanographic applications. The Laplacian of the directed

graph G is defined, Chung (2005),

$$\mathcal{L} = I - \frac{\Pi^{1/2}\mathcal{P}\Pi^{-1/2} + \Pi^{-1/2}\mathcal{P}^T\Pi^{1/2}}{2}. \tag{12}$$

The first smallest eigenvalue larger than zero, $\lambda_2 > 0$ such that,

$$\mathcal{L}v_2 = \lambda_2 v_2, \tag{13}$$

allows a bi-partition, by the sign structure of,

$$y = \Pi^{-1/2}v_2. \tag{14}$$

Analogously to the Ng-Jordan-Weiss symmetric spectral image partition method Ng et al. (2002), the first $k$ eigenvalues larger than zero, and their eigenvectors, can be used to associate a multi-part partition, by the assistance of $k$-means clustering of these eigenvectors. By defining the matrix $V = [v_1, v_2, \ldots, v_k]$, that have the eigenvectors associated with the $k^{th}$ most significant eigenvalues on its columns, then we use the $k$-means clustering to multi partition $V$ based on the $L_2$ distance between $V$'s

rows. Since each row in the matrix $V$ is associated with a specific spatial location (pixel), then by reshaping the labels vector that results from the $k$-means clustering, we obtain our labeled image.

## 3 Results

We apply the Ddirected Affinity Segmentation to satellite images of the Larsen C ice shelf and ice surface velocity data. Here we show that the DAS of spatiotemporal changes can work as an early warning sign tool for critical transitions in marine

ice sheets. We applied our "post-casting" experiments on Larsen C images before the splitting of the A68 iceberg, then we compared our forecasting based on segmentation to the actual unfolding of the event.

In Fig. 3, we see different snapshots of the ice surface velocity data set E. Rignot, J. Mouginot and B. Scheuchl (2017); Rignot et al. (2011); Mouginot et al. (2012), which are part of the NASA Earth system data records for use in the research environments (MEaSUREs) program. It provides the first comprehensive E. Rignot, J. Mouginot and B. Scheuchl (2017), high-

180 resolution, digital mosaics of ice motion in Antarctica assembled from multiple satellite interferometric synthetic-aperture radar systems. We apply our directed affinity partitioning algorithm to these available data sets, and the results are shown as a labeled image in Fig. 4.

As shown in Fig. 4, we note the following:

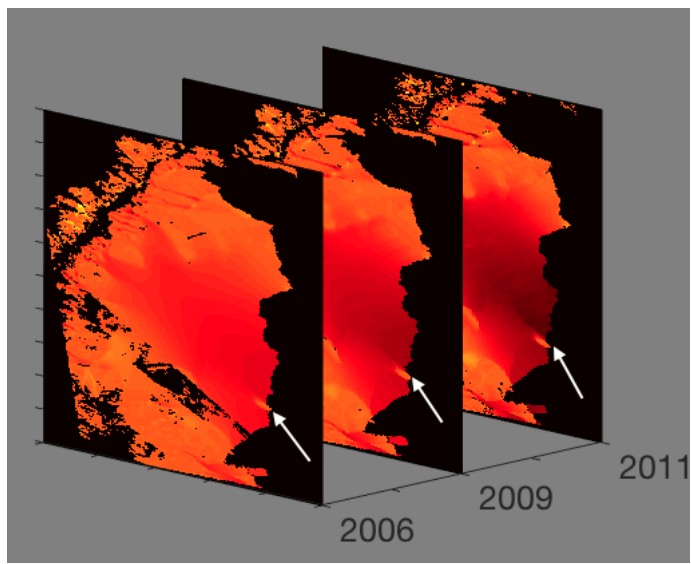

**Figure 3.** Ice surface velocity. The figure shows the data set for three different years around the beginning of the Larsen C ice crack in 2010. The data from the years 2007, 2008 and 2010 are corrupted on the region of interest, and they are excluded. The color scale indicates the magnitude of the velocity from light red (low velocity) to dark red (high velocity), and the arrow points to the starting tip of the crack. The result of the directed partitioning is shown in Fig. 4. Source of data: E. Rignot, J. Mouginot and B. Scheuchl (2017).

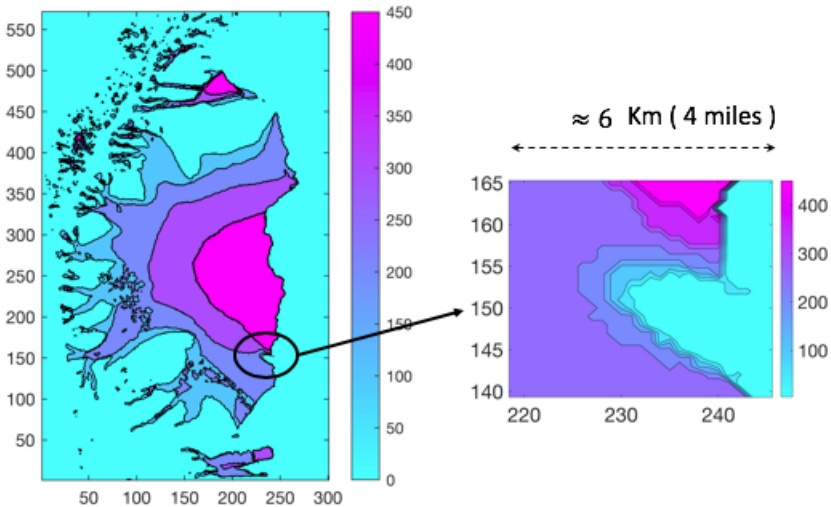

**Figure 4.** Directed Affinity result. (Left) The directed partitioning results for the ice surface velocity of 2006, 2009, 2011, and 2012. Note that the ice shelf crack started in 2010. (Right) A narrow field zoom to the region of interest shows large variations of ice surface velocity within a small area, to give a clearer focused view of the differences in speeds. In Appendix, Fig. B1 shows the surface plot for the same result.

- The data was collected from eight different sources E. Rignot, J. Mouginot and B. Scheuchl (2017); Map (2017), with different coverage and various error ranges, and interpolating the data from these different sources explains the smooth curves in segmentation around the region of interest.

- The directed partitioning shows the Larsen C ice shelf as a nested set of coherent structures that are contained successively within each other.

- The zoom scene shown in the right of Fig. 4 highlights the region where the Larsen C ice crack starts. Furthermore, we see a significant change of velocity within a narrow spatial distance (4 miles). More precisely, the outer boundaries of coherent sets become spatially very close (considering the margin of error in the measurements Map (2017). We conclude that, likely, these contact).

Directed partitioning gives us informative clustering, meaning that each cluster has homogeneous properties, such as the magnitude and the direction of the velocity. Consider the nested coherent sets, $A_1 \subset A_2 \subset ... \subset A_n$, shown in Fig. 5. Each set $A_{i-1}$ maintains its coherence within $A_i$ because of a set of properties (i.e., chemical or mechanical properties) that rules the interaction between them. However, observe that the contact between the boundaries of the sets $A_{i-1}$ and $A_i$, can mean a direct interaction between dissimilar domains. These later sets may significantly differ in their properties, such as a significant difference of velocity, which may require different analysis under different assumptions than the gradual increase in the velocity.

However, since the sets' boundaries are not entirely contacted, the velocities' directions reveal no critical changes; we believe this results implicitly from the data preprocessing that includes interpolation and smoothing of the measurements. We believe that the interpolation and smoothing of the measurements cause loss of data informativity about critical transitions. Our method, using the ice surface velocity data, was able to detect more details. However, it still cannot detect critical transitions such as the crack branching, as discussed in the introduction, and shown as in Fig. A2. Based on our results using the ice velocity data, we state nothing more than such close interaction between coherent sets boundaries. As shown in Fig. 4, an early warning sign should be considered and investigated by applying the potential hypothesis ("what if" assumptions) and analyzing the consequences from any change or any error in the measured data.

It is interesting to contrast our directed partitioning results, which give early indications of impending fracture changes using the remote sensing satellite images, to classical interferometry analysis methods Scambos et al. (1996). To reduce the obscuration effects of noise (clouds and image variable intensity), we used the averaged images, over one month, as a single snapshot for the directed affinity constructions. We excluded some images that have high noise and lack of clarity in the region of interest. See Fig. B8. Fig. 6, the directed affinity partitioning for two time-windows starts from December 2015. Notice that the directed partitioning begins to detect the Larsen C ice shelf's significant change in July 2016. In Fig. 7, we see that by September 2016, we detect a structure very close in shape to the eventual and actual iceberg A-68, which calved from Larsen C in July 2017. Moreover, by November 2016, see Fig. B2, the boundaries of the detected partitions match the crack dividing into two branches that happened later in May 2017, and shown in Fig. A2.

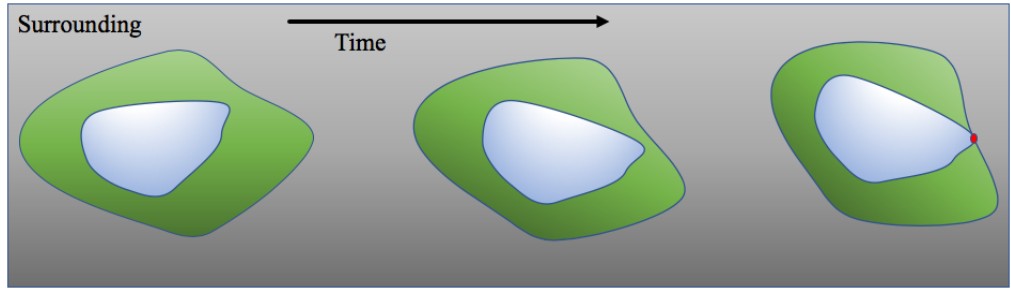

**Figure 5.** Two coherent sets dynamic. As the inner set contacts the boundary of the outer one, it gives the chance for a new reactions that "may" cause a critical transition.

## 4   Discussion

We have shown that our data-driven approach, originally developed for detecting coherent sets in fluidic systems, shows promise for predicting possible critical transitions in spatiotemporal systems, specifically for marine ice sheets, based on remote sensing satellite imagery. Our approach shows reliability in detecting coherent structures, when the object of concern is a quasi-rigid body such as ice sheets. The main idea is that observing a significant and perhaps topological form change of a coherent structure may indicate an essential underlying critical structural change of the ice over time. The computational approach is based on spectral graph theory in terms of the directed graph Laplacian. We have shown here that carefully designing a directed affinity matrix, which accounts for balancing spatial distance, and measurements at spatial sites, for application of spectral graph theory, is relevant in our applied setting of remote sensing imagry. In the case of the Larsen C ice shelf, we have carried forward this data-driven program. We successfully observe the calving event of the A68 iceberg and some critical transitions months before their actual occurrence. This transition of the coherent structure can indicate a possible fracture, by directed affinity partitioning. We see that the directed affinity partitioning can be a useful early warning sign that indicates the possibility of critical spatiotemporal transitions, and it may help to bring the attention to specific regions to investigate different possible scenarios in the analytic study, whether these be further computational analysis or possibly even supporting further field studies and deployed aerial remote sensing missions. We have demonstrated in the case of the Larcen C ice shelf event, with evidence of Figs 6-8 and B1-7 that potentially important events may be observable months ahead of the final outcome.

In our future work, we plan to pursue the idea of connecting our data-driven approach of computing boundaries by directed partitioning with the computational science approach in terms of stress/strain analysis of rigid bodies and an understanding of the underlying physics.

*Data availability.*   The data of this study are available from the authors upon reasonable request.

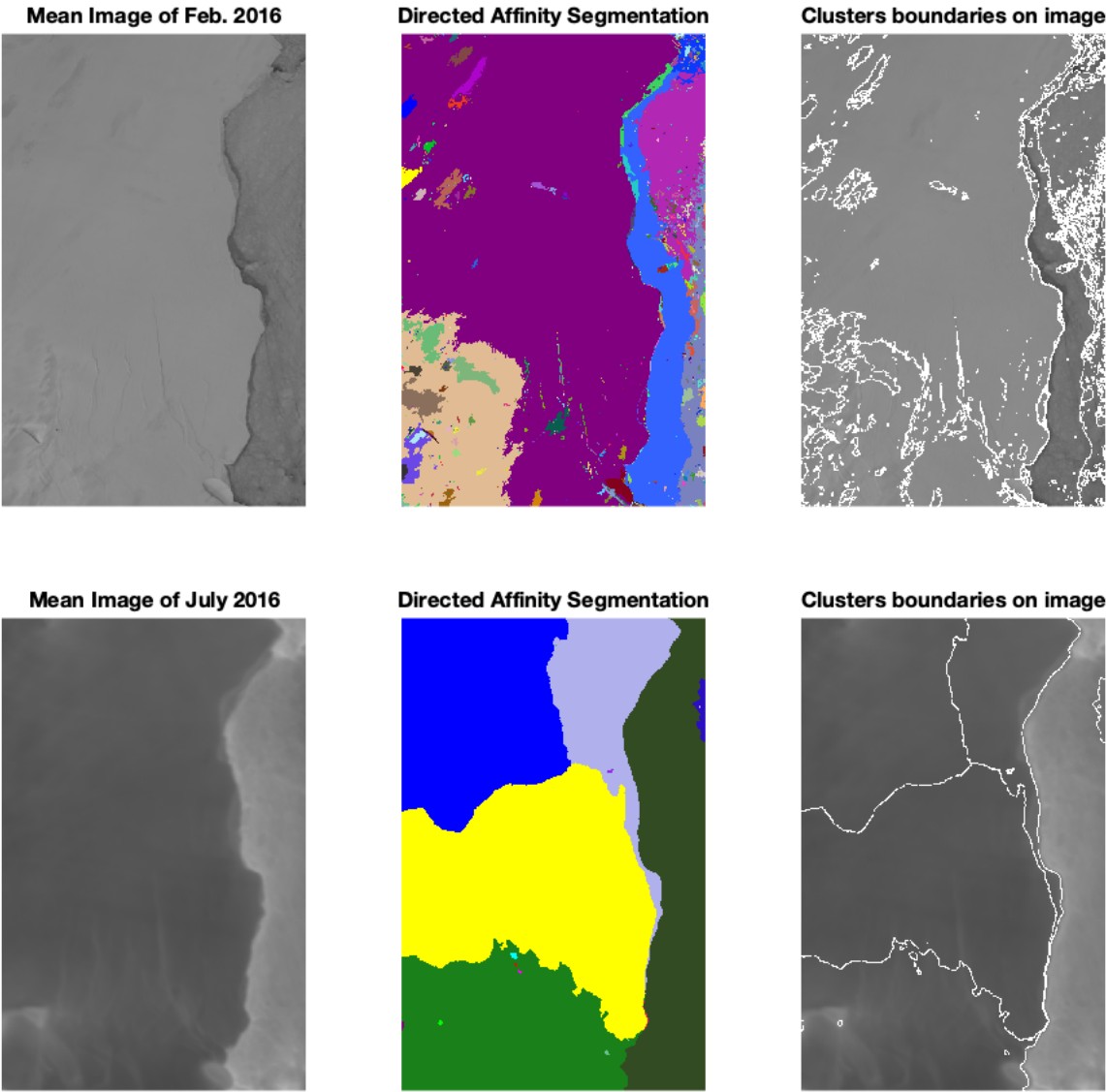

**Figure 6.** For two time-windows (top and bottom), we see (Left) The mean image of the images included in the window. (Middle) The Directed Affinity Segmentation labeled clusters. (Right) Overlay the directed affinity segmentation boundaries over the mean image of the window. We took these two time windows of Feb. 2016 and July 2016 as a detailed example, and more time windows results are shown in Fig. 7. During 2016, there was no significant change in the Larsen C crack at the beginning of the year. However, in July 2016, based solely on data up to that point in time, the directed affinity segmentation propose a large change in the crack dynamics, and this change the continues faster as Fig. 7 shows. Raw images source Scambos et al. (1996).

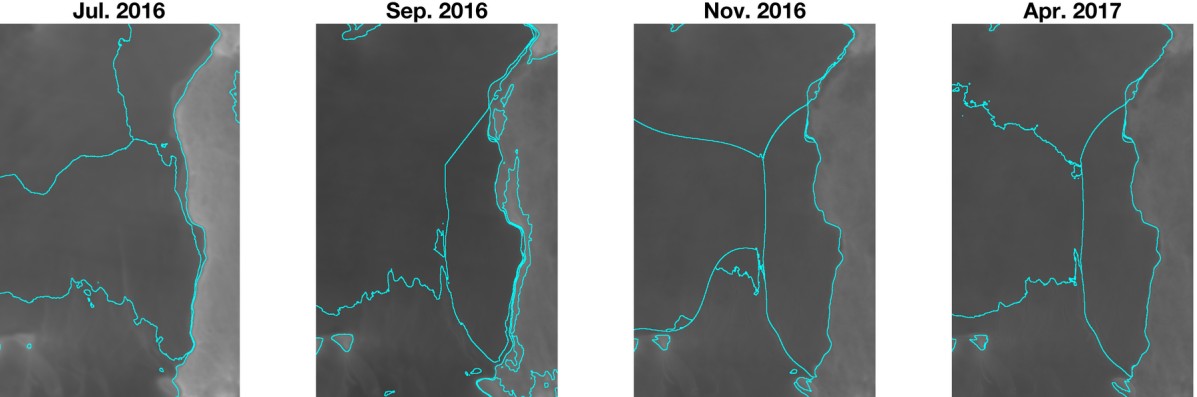

**Figure 7.** In analogy to Fig. 6-Right, this figure shows the Directed Affinity Segmentation boundaries for different time windows starting from July 2016 to April 2017. Raw images source Scambos et al. (1996).

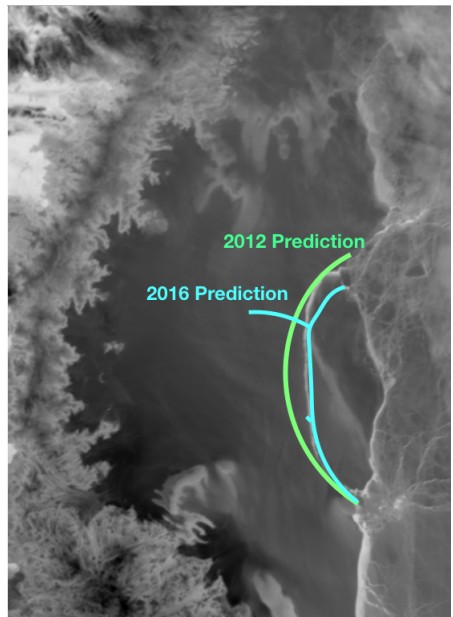

**Figure 8.** 2012 prediction based on ice surface velocity data, and 2016 prediction based only on satellite images. Compare to the actual crack (white curve between the two prediction curves) on July 2017, shown in Fig. 1. Raw image source Scambos et al. (1996).

## Appendix A: Figures

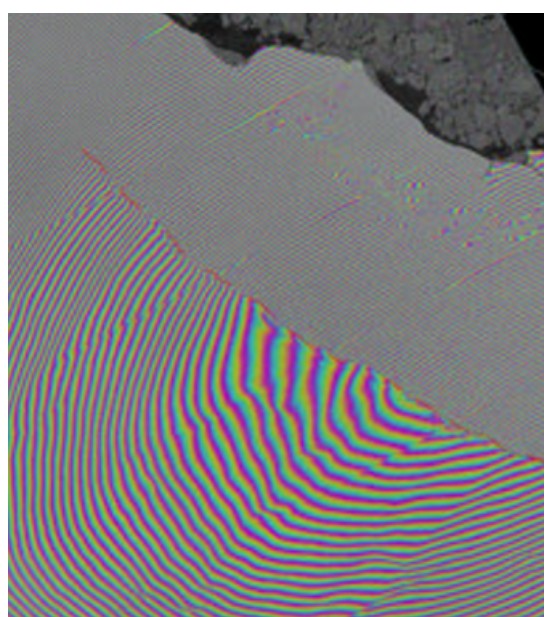

**Figure A1.** Interferometry (April 20, 2017). Two Sentinel-1 radar images from 7 and 14 April 2017 were combined to create this interferogram showing the growing crack in Antarctica's Larsen-C ice shelf. Polar scientist Anna Hogg said: "We can measure the iceberg crack propagation much more accurately when using the precise surface deformation information from an interferogram like this, rather than the amplitude (or black and white image) alone where the crack may not always be visible." Source Agency (2017).

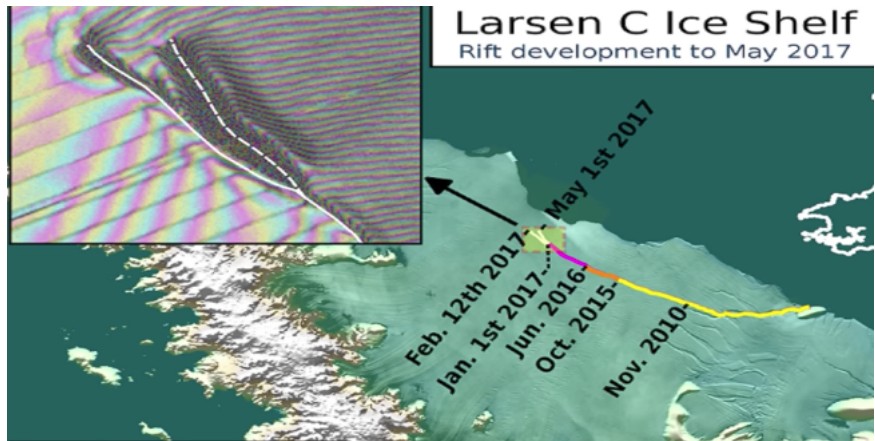

**Figure A2.** Lrasen C crack development (new branch) as of May 1, 2017. Labels highlight significant jumps. Tip positions are derived from Landsat (USGS) and Sentinel-1 InSAR (ESA) data. Background image blends BEDMAP2 Elevation (BAS) with MODIS MOA2009 Image mosaic (NSIDC). Other data from SCAR ADD and OSM. Credit: MIDAS project, A. Luckman, Swansea University.

## Appendix B: More Numerical Results

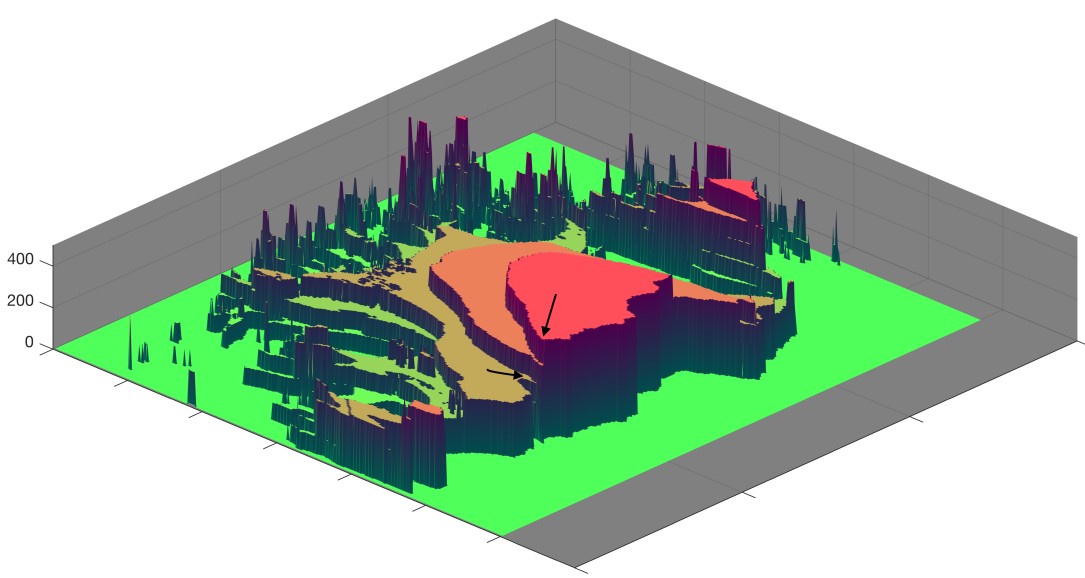

**Figure B1.** Directed affinity partitions with the mean velocity (speed) of the partition assigned for each label entries. The spatial distance between the arrows tips is less than two miles, while the difference in the speed is more than 200 m/year.

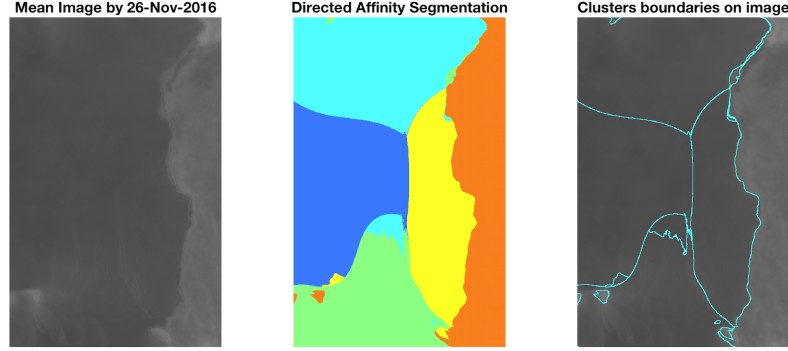

**Figure B2.** The mean image and the directed affinity partitioning as of November 2016. The results shows similar structure to the crack branching that occurred on May 2017 and shown in Fig. A2, and similar structure the final iceberge that calved from Larsen C on July 2017. Raw images source Scambos et al. (1996).

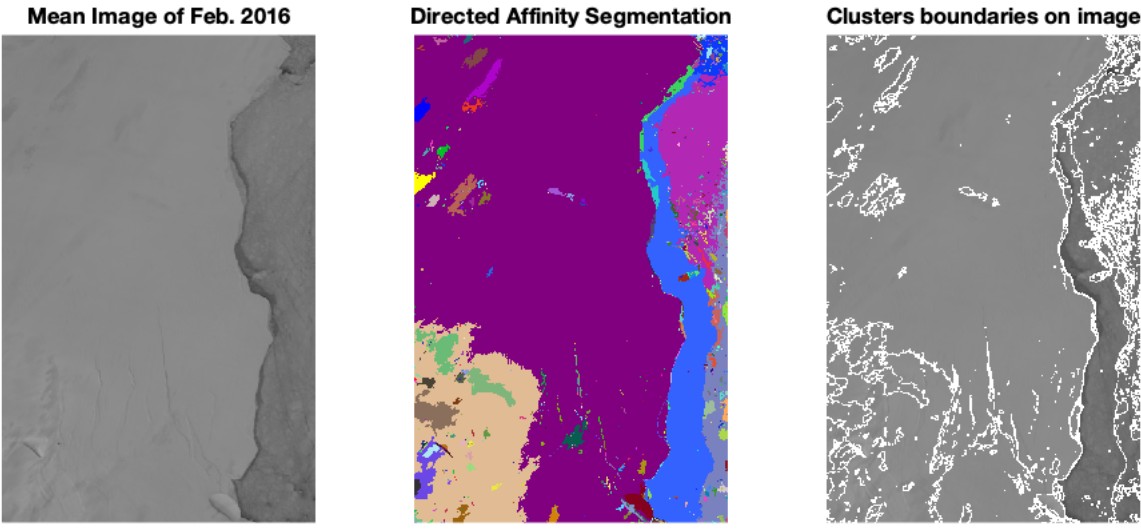

**Figure B3.** The mean image and the directed affinity partitioning as of February 2016. Raw images source Scambos et al. (1996).

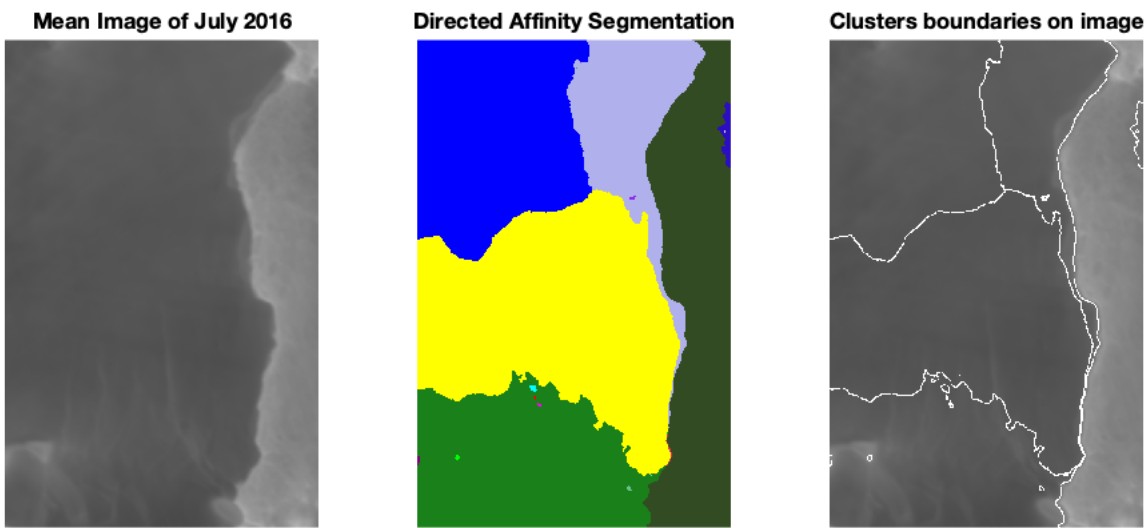

**Figure B4.** The mean image and the directed affinity partitioning as of July 2016. Raw images source Scambos et al. (1996).

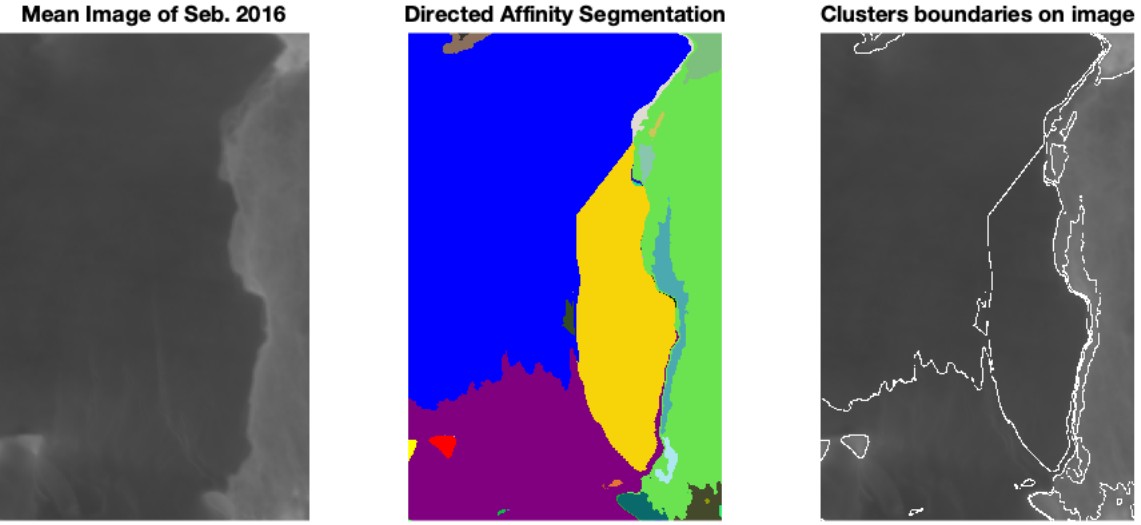

**Figure B5.** The mean image and the directed affinity partitioning as of September 2016. Raw images source Scambos et al. (1996).

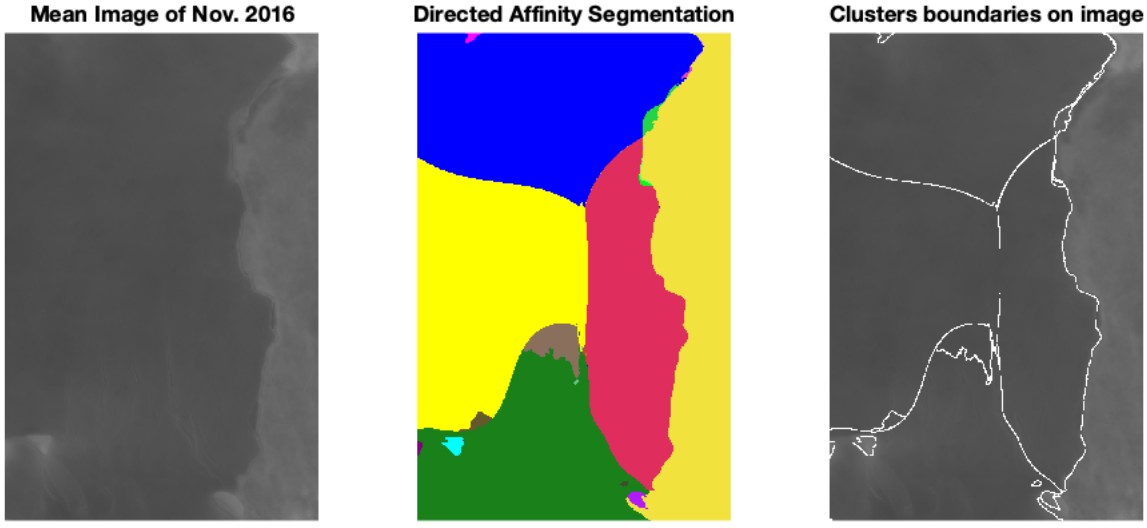

**Figure B6.** The mean image and the directed affinity partitioning as of November 2016. Raw images source Scambos et al. (1996).

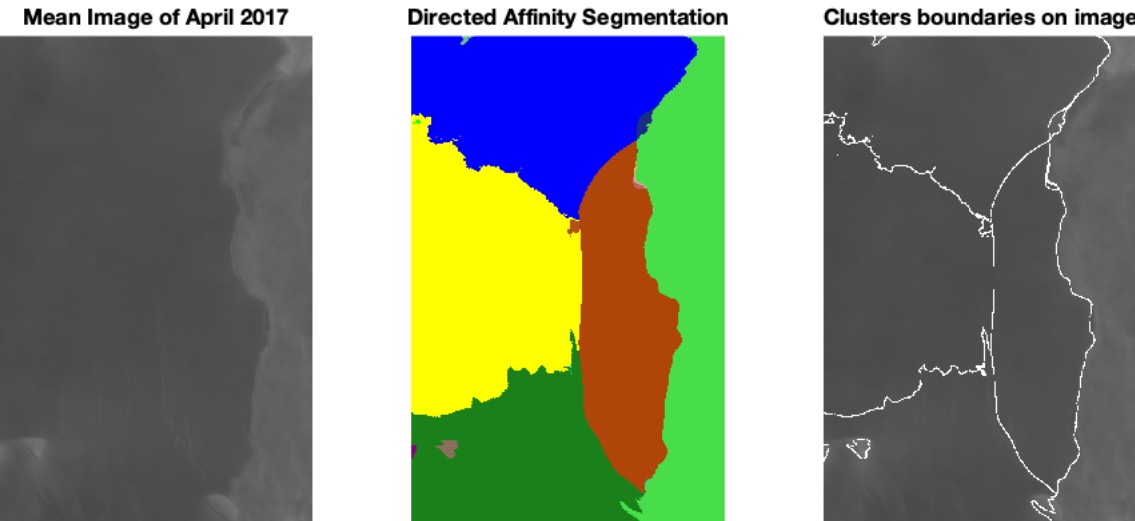

**Figure B7.** The mean image and the directed affinity partitioning as of April 2017. Raw images source Scambos et al. (1996).

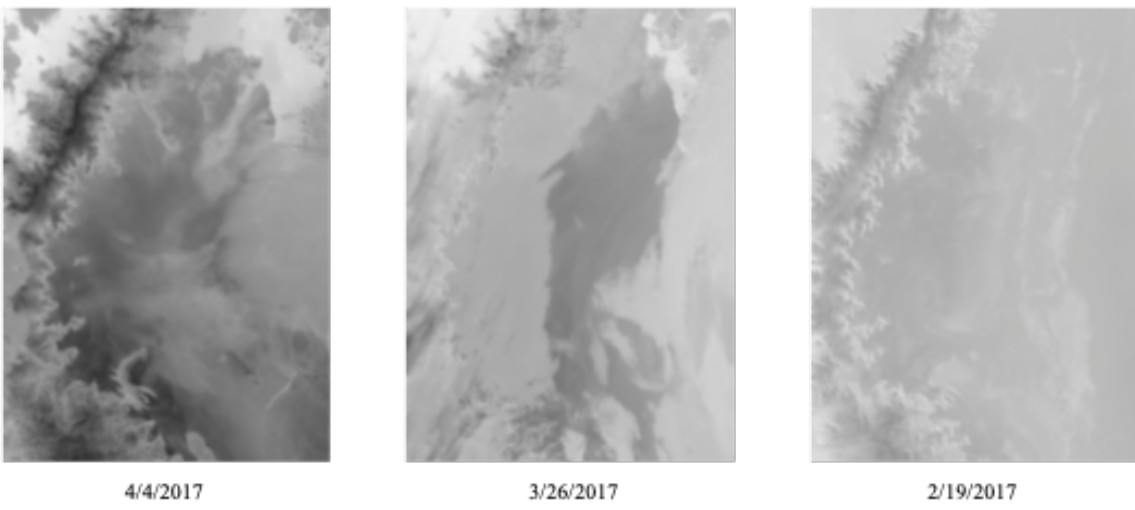

**Figure B8.** Example of noisy images that have been excluded when computing the average image. Raw images source Scambos et al. (1996)

*Author contributions.* Each author contributed approximately 50% time effort to this project. EB developed conceived the background theory, was involved in the design of experiments and wrote much of the manuscript. AA implemented theory, developed the specific experimental design, codes, and analysis, and also wrote much of the manuscript.


*Competing interests.* The authors declare no competing interests.

*Acknowledgements.* This work was funded in part by the Army Research Office, the Naval Research Office, the Army Research Office, and also DARPA.

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
