# Peer review of "An Early Warning Sign of Critical Transition in The Antarctic Ice Sheet - A Data Driven Tool for Spatiotemporal Tipping Point"

_Nonlinear Processes in Geophysics, 2020_

## Referee Comment (RC1) · Anonymous Referee #1 · 8 Sep 2020

This paper presents a data-driven methodology for detecting early-warning signs of critical transitions on ice sheets. The approach is based on a spectral partitioning of image data acquired by remote sensing, using a directed graph equipped with an asymmetric affinity matrix constructed from lagged sequences of images. The method is applied to ice surface velocity data for the Antarctic, and is found to successfully detect the formation of the A68 iceberg in the Larsen C ice shelf that took place in 2017.

Overall, my assessment is that this is an interesting paper, worthy of publication at NPG. I recommend revisions to clarify some aspects of the analysis and improve presentation, as detailed below.

1. The introduction, as well as the conclusions, read overly critical of interferometric approaches as a tool for analysis and prediction of sea ice cracks. I wonder, however, if the issue here is not with interferometry itself but rather with how the data is processed in order to extract information pertinent to crack formation. After all, as stated in lines 169–175, the velocity data utilized in this study are at least partly based on interferometry, so whatever information the proposed methodology extracts was at least partially present in interferometric data.

2. Section 2 describes the graph affinity matrix as being constructed from color data, but the text in lines 169–175 suggests that ice surface was used. Please clarify and explicitly state the data sources employed in the analysis.

3. Although I believe that this is the case, it is not fully clear whether the results in figures 4, 7, and elsewhere in the paper are predictive in nature. That is, if the directed partitioning method detects significant changes in July 2016, is this based solely on data up to that point in time? It would be helpful to explicitly state this.

4. What is the sensitivity of the results on $\tau$, $\alpha$, and $\sigma$ parameters in the graph affinity function? In general, there is little information about how these parameters are chosen. Similarly, other than a high-level reference to $k$-means clustering, there is little information about how the eigenvectors of the graph Laplacian are employed to produce the final image segmentation. These issues considerably affect the reproducibility of the results, and it is important that the implementation of the technique is adequately explained in the revised manuscript.

5. Consider rewording the sentence in lines 189-191 (describing the partitions $A_j$) as it appears to be grammatically incorrect. Similarly the text in lines 194-200 could be improved in terms of English/clarity.

2020-26, 2020.
Interactive
comment

---

## Referee Comment (RC2) · Anonymous Referee #2 · 6 Oct 2020

Please note, I am a geophysist who considered the glaciology and mechanics in this paper. I do not comment on the mathematical method. In that context I would like to say it is exciting to see new mathematical methods to extract discontinuities in velocity field in glacial ice. It is interesting that one can estimate the onset of the crack formation, and perhaps with subsequent images the crack propagation. I did not assess if the method is able to show the velocity discontinuity within measurement error, but if it is a real result the method should be of interest to the cryospheric community.

Specific points

line 22: "Still, this contribution starts to change in the 21st century because of the ice

shelves cracks". This sentence is rather clunky. Ice shelf retreat? Or increased iceberg calving?

There are other places with clunky English. For example line 35 "attribute in Greenland" is not grammatically correct. r line 55 "most massive known iceberg" is not formal language. I would suggest having someone proof read for professional English who is in the field.

paragraph 37-42: Not sure if this is needed. It is a little out of context. There are other examples of information that is interesting but is out of context of the immediate point of interest, ice shelf cracking.e.g. "Interestingly, two and a half years later, it remains mostly intact and has drifted from the near Antarctica seas into the more turbulent open Arctic Ocean where it is expected to break apart more quickly." .... I would suggest a proof read focused on direct narrative in the paper.

In general the introduction could be more focused to ice shelf processes that involve it's growth and ice loss through iceberg generation.

There are spelling mistakes in the manuscript

---

## Author Comment (AC1) · 21 Oct 2020

We thank the anonymous reviewers for their careful reading of our manuscript and their insightful comments and suggestions. Following the suggestions, we included several improvements in the manuscript resulting in a stronger and clearer manuscript. Below, we will give a detailed replay to the comments.

Anonymous Referee #1:

"This paper presents a data-driven methodology for detecting early-warning signs of critical transitions on ice sheets. The approach is based on a spectral partitioning

of image data acquired by remote sensing, using a directed graph equipped with an asymmetric affinity matrix constructed from lagged sequences of images. The method is applied to ice surface velocity data for the Antarctic, and is found to successfully detect the formation of the A68 iceberg in the Larsen C ice shelf that took place in 2017. Overall, my assessment is that this is an interesting paper, worthy of publication at NPG. I recommend revisions to clarify some aspects of the analysis and improve presentation, as detailed below."

We want to thank the referee for his careful read, positive feedback, and constructive comments.

1. The introduction, as well as the conclusions, read overly critical of interferometric approaches as a tool for analysis and prediction of sea ice cracks. I wonder, however, if the issue here is not with interferometry itself but rather with how the data is processed in order to extract information pertinent to crack formation. After all, as stated in lines 169–175, the velocity data utilized in this study are at least partly based on interferometry, so whatever information the proposed methodology extracts was at least partially present in interferometric data.

We agree with the reviewer that "the issue here is not with interferometry itself but rather with how the data is processed in order to extract information pertinent to crack formation". We changed several sentences to reflect this tone, and we updated the manuscript to reflect more clarifications about the comparison. See the revised manuscript, line 206-222. We clarified that using the ice velocity data, our method revealed interesting details. Still, it could not predict the critical change and branching of the crack that happened in May 2017. On the other hand, using only the satellite images, our method was able to detect this critical branching by November 2016, and it was able to predict more accurate boundaries to the overall calved iceberg.

2. Section 2 describes the graph affinity matrix as being constructed from color data, but the text in lines 169–175 suggests that ice surface was used. Please clarify and

explicitly state the data sources employed in the analysis.

We thank the reviewer for the comment on this important point. In lines 116-121, added a discussion to clarify that our method is not limited to a specific measured quantity, and we state that: "It is crucial to keep in mind that we chose the color as the evolving quantity for a designated spatial location for clarity and consistency with our primary application and approach introduced in this paper. However, we can select the evolving quantity to be the magnitude of the pixels obtained from spectral imaging or experimental measures obtained from the field, such as pressure, density, or velocity. The results section introduces examples where we used the ice surface velocity instead of the color to show how results may vary based on the selected measure".

In the results section, we ensured that the data source is cited clearly in each figure caption.

3. Although I believe that this is the case, it is not fully clear whether the results in figures 4, 7, and elsewhere in the paper are predictive in nature. That is, if the directed partitioning method detects significant changes in July 2016, is this based solely on data up to that point in time? It would be helpful to explicitly state this.

We updated the document, and we emphasized this point in the discussion on the caption of Figure 5, and we clarified that the results were based solely on data up to that point in time.

4. What is the sensitivity of the results on $\tau$, $\alpha$, and $\sigma$ parameters in the graph affinity function? In general, there is little information about how these parameters are chosen. Similarly, other than a high-level reference to k-means clustering, there is little information about how the eigenvectors of the graph Laplacian are employed to produce the final image segmentation. These issues considerably affect the reproducibility of the results, and it is important that the implementation of the technique is adequately explained in the revised manuscript.

In lines 110-114 and lines 126-131, we added a discussion on the parameters' sensitivity and selection. In lines 167-171, we added a paragraph that clarifies the main principle in applying the K-means clustering on the graph Laplacian's eigenvectors and how we obtain our labeled image.

5. Consider rewording the sentence in lines 189-191 (describing the partitions Aj) as it appears to be grammatically incorrect. Similarly the text in lines 194-200 could be improved in terms of English/clarity.

We revised the sentence in lines 189-191 and reworded it for more clarity. You can see the revised paragraph in lines 196-202. Also, we carried an extensive review throughout the manuscript, for clarity, English, and grammatical errors.

Anonymous Referee #2:

"Please note, I am a geophysist who considered the glaciology and mechanics in this paper. I do not comment on the mathematical method. In that context I would like to say it is exciting to see new mathematical methods to extract discontinuities in velocity field in glacial ice. It is interesting that one can estimate the onset of the crack formation, and perhaps with subsequent images the crack propagation. I did not assess if the method is able to show the velocity discontinuity within measurement error, but if it is a real result the method should be of interest to the cryospheric community."

We want to thank the referee for his positive feedback and constructive comments.

Specific points:

1. line 22: "Still, this contribution starts to change in the 21st century because of the ice shelves cracks". This sentence is rather clunky. Ice shelf retreat? Or increased iceberg calving? There are other places with clunky English. For example line 35 "attribute in Greenland" is not grammatically correct. r line 55 "most massive known iceberg" is not formal language. I would suggest having someone proof read for professional English who is in the field.

We thank the reviewer for his careful read and helpful comments. We revised the mentioned sentences and marked them in blue in our revised manuscript. And we carried an extensive review all over the manuscript, for clarity, English, and grammatical mistakes.

2. paragraph 37-42: Not sure if this is needed. It is a little out of context. There are other examples of information that is interesting but is out of context of the immediate point of interest, ice shelf cracking.e.g. "Interestingly, two and a half years later, it remains mostly intact and has drifted from the near Antarctica seas into the more turbulent open Arctic Ocean where it is expected to break apart more quickly." .... I would suggest a proof read focused on direct narrative in the paper. In general the introduction could be more focused to ice shelf processes that involve it's growth and ice loss through iceberg generation.

We thank the reviewer for his careful read and helpful comments. We agree with the reviewer, and we removed the mentioned sentences, with several other sentences all through the manuscript to focus on our main objectives and subject.

3. There are spelling mistakes in the manuscript

We carried an extensive review all over the manuscript for spelling and grammatical mistakes.

Please also note the supplement to this comment:
https://npg.copernicus.org/preprints/npg-2020-26/npg-2020-26-AC1-supplement.pdf

---

## Author Response (AR2)

Dear Editor,

Thank you for your careful reading of our manuscript and your insightful comments. Following the comments and suggestions, we have included several changes in the manuscript. Below, we give a detailed reply to the comments.

it states that it that the methodology is introduces...new" yet there's a reference to published work in which the method is developed. It is thus not new.

We re-wrote the sentence, and now it reads, "This paper newly introduces that the use of our recently developed tool, that was originally designed for data-driven discovery of coherent sets in fluidic systems, can in fact be used to indicate early warning signs of critical transitions in ice shelves, from remote sensing data."

"Our approach adopts...considering..." I am not sure "considering" is the right word...

Our approach adopts a directed spectral clustering methodology in terms of developing an asymmetric affinity matrix and the associated directed graph Laplacian.

"We generally apply" as opposed to specifically apply??

Thank you. we replaced "generally" with "specifically."

"(post-cast) predict" do you mean forecast?

Thank you very much. We added the following clarification for the term "post-cast": (such benchmarking using data from the past to forecast events that are now also in the past is sometimes called ``post-casting," analogously to forecasting into the future)

"We can do so months earlier before" rewrite

Thank you. The sentence now reads: "Our method indicates the coming crisis months before the actual occurrence, and furthermore, much earlier than any other previously available methodology, particularly those based on interferometry".

The discussion on warming is not properly framing either the problem you wish to consider or how warming and calving and other fracture events connect with each other. I think you are trying to say that with warming there are more sheet-cracking events...
but the reader is left with the question: 'I can see how warming might have something to do with cracking' but is this fundamental to the methodology to be presented? is warming a requisite for the methodology to be applicable?

Thank you very much for the insightful comment. The discussion on warming is highly reduced to two sentences, and considers your comment.

"Still, this contribution..." I don't understand the use of "Still"
Reference to calving and the change in the 21st century?

We re-wrote the sentence and added a reference to calving and the change in the 21st century.

l26-31 "Most of Antarctica...dramatic warming" then it says that it has warmed 2.5degC...in my book this is dramatic, is it not?

Thank you. Yes, it is a dramatic warming. We re-wrote the sentence so that now it states: "Antarctica already experienced dramatic warming. Especially, the Antarctic Peninsula, which juts out into relatively warmer waters north of Antarctica, has warmed 2.5 degrees Celsius (4.5 degrees Fahrenheit) since 1950"

l32 "probably due to" either it is or not or there's contradicting or incomplete evidence for this...as it reads, it sounds like it is your opinion.

Thank you. The sentence now reads: "A large area of the Western Antarctic Ice Sheet is also losing mass, attributed  to warmer water up-welling from the deeper ocean near the Antarctic coast."

Finally when it gets to methodologies it mentions interferometry...fine, what's wrong with this method? instead of stating you need to contrast your method with interferometry, why don't you actually contrast them? and further, can't you construct an extrapolation based upon interferometric data to suggest what happens next?

Thank you very much. We have reviewed and modified our discussion to better present our point of view. The main perspective is that interferometry is clearly an excellent and powerful tool that is well suited to the job of detecting ice surface velocity. However, it is not specifically a "prediction" method in of itself. It is a tool that provides information that can be used as a starting point to apply further and different types of analysis that can conclude some "predictions." On the other hand, applying our method, which is originally designed for detecting and tracking coherent structures, to a rigid body can provide different insights and give an early indication of the possible critical transitions. See line 82-85.

showed "high performance in successfully detecting" meaning what? less false positives, more positives, etc? how was this vetted? and what are "fluidic systems" (and do fluids have cracks, are we talking about multiphase flow?)

We re-wrote the sentence, so that now it reads: ``We showed that our method is analytically a data-driven analogue to the transfer operator formalism designed for detecting, which was originally designed for coherent structures in fluidic systems, such as ocean flows or atmospheric storms.''

l 91 "what inderlies a notion...we must understand directionality" this sentence is muddled

Thank you. We re-wrote the sentence, and now it reads: ``The key difference is what underlies a notion of coherent observations that we must also consider the directionality of the arrow of time.''

'The rest of this section needs a serious, sentence by sentence analysis and revamping so that the grammar is up to basic standards for publication. In fact, the whole paper needs this treatment.

Thank you. We have taken this to heart. We have continued to carefully edit the text throughout. Some of which are here, but much of it is minor but numerous improvements.

how are you quantifying color and what is the resolution? Cal S and Cal C could be radically different in sizes, so there must be some sort of normalization...is there?
The parameter alpha also has to be accounted for in this regard.

Thank you. We clarify the quantification and the scale of S and C. We also provide a discussion about the effect of the alpha value on the results. See lines 137-143.

The results have very little by way of quantitative analysis. You declare the method superior, but I do not see evidence for this, or it is rather circumstantial, at best.
If this section had more compelling outcomes the reader might be incentivized to try it.

The referee's point is well taken, that statements of a superior method suggest that a statistical analysis would follow to show what it is the fractional improvement over the other methods. In this case, we do not know of other forecasting methods, and even our method is not necessarily a forecasting method. However, we do show that this important event showed itself from the perspective of our data driven approach, months in advance, and we know of no other comparison for this striking observation. So, we hope this pictorial evidence, Figs. 6-8, and B1-B7 processed in the manner of the presented algorithm, in of itself, serves as a demonstration utility to detect this rare event.

There's a serious missed opportunity to discuss the color issue. Surely, there are "better" color than others. In any event, I am not expecting that this issue will be addressed here, but I will suggest doing so in order to produce a more compelling paper.

Thank you very much. Yes, we agree that it is interesting issue to address and discuss, and it will be a subject for our future work.

Section 4

one very obvious issue to address is the role of noise in the data. I can imagine that if the data is not prepared properly that noise will lead to a lot of false details. You averaged the data over a month...is there some quantitative guidance for data preparation?

Thank you. In line 225-228, we discussed the effect of noise, and how we excluded some noisy images because of their effect on the results. Also, we added a new figure, Fig.B.8, that shows a sample of the noisy images that was excluded.

The idea of connecting mechanics to this data-driven method is sensible, but I am not sure I would agree that it'd be productive to consider an asymptotic-in-time method like Lyapunov vectors and exponents with something that seems to be designed for short time events? At the very least you'll need to explain yourself here (or omit).

Thank you very much. You are right. We said too little for this to be meaningful, and we decided it would take too much space to clearly make this otherwise auxiliary comment. So, we decided to take your advice and omit the comment.

Sincerely,

Abd AlRahman AlMomani and Erik Bollt

---

## Author Response (AR3)

Dear Editor,

Thank you for your continued feedback concerning our manuscript, npg-2020-26

Title: An Early Warning Sign of Critical Transition in The Antarctic Ice Sheet -A New Data Driven Tool for Spatiotemporal Tipping Point
Author(s): Abd AlRahman AlMomani and Erik Bollt
MS No.: npg-2020-26
MS type: Research article
Iteration: Minor Revision

We greatly appreciate the time and effort by you and the referees.

We have worked to address each of the concerns.  We believe that all the suggestions have been excellent and that adjusting for these as best we could, that the improved manuscript should be ready for final acceptance.  We hope the editor agrees.

Sincerely,
Erik Bollt,
Abd Al Rahman Al Momani

Enumerated, the changes were,

1) English.  We have carefully worked through the entire manuscript from title, and abstract, through to the conclusions.  We have made extensive minor adjustments throughout.  So many, that the relevance of coloring in blue every change looses relevance.  I am sure if you read any part, you will now find that the text is cleaner and smoother we hope.  We have colored more central changes, especially including the title, the abstract, and in the conclusion, but this is not meant to be exclusive to the many other changes.
2) We have now changed the title and abstract as suggested.

Title: An Early Warning Sign of Critical Transition in the Antarctic Ice Sheet - A Data Driven Tool for Spatiotemporal Tipping Point

Abstract: Our recently developed tool, called Directed Affinity Segmentation was originally designed for data-driven discovery of coherent sets in fluidic systems.  Here we interpret that it can also be used to indicate early warning signs of critical transitions in ice shelves, as seen from remote sensing data. We apply a directed spectral clustering methodology, including an asymmetric affinity matrix and the associated directed graph Laplacian, to reprocess the ice velocity data and remote sensing satellite images of the Larsen C ice shelf. Our tool has enabled the simulated prediction of historical events from historical data,  fault lines responsible for the critical transitions leading to the breakup of the Larsen C ice shelf crack, which resulted in the A68 iceberg. Such benchmarking of methods using data from the past to forecast events that are now also in the past is sometimes called post-casting, analogous to forecasting into the future. Our method indicated the coming crisis months before the actual occurrence.

3) We have specifically altered the language to weaken the overly strong claim pointed out by the referee, and several others.  See the new text colored in blue, lines 226-227:

It is interesting to contrast our directed partitioning   results, which give early indications of impending fracture changes using the remote sensing satellite images, to classical interferometry analysis methods Scambos et.al. (1996).